# In Situ Simulation Training for Frailty

**DOI:** 10.3390/geriatrics8010026

**Published:** 2023-02-17

**Authors:** Liam Dunnell, Anna Nicole Barnard, Katie Chu, Ania Barling, Jonathan Birns, Grace Walker

**Affiliations:** 1Simulation and Interactive Learning Centre, Guy’s and St Thomas’ NHS Foundation Trust, London SE1 7EH, UK; 2Department of Ageing and Health, Guy’s and St Thomas’ NHS Foundation Trust, London SE1 7EH, UK

**Keywords:** simulation, training, education, frailty, interprofessional

## Abstract

Background: People living with frailty account for a significant proportion of hospital inpatients and are at increased risk of adverse events during admission. The understanding of frailty remains variable among hospital staff, and there is a need for effective frailty training across multidisciplinary teams. Simulation is known to be advantageous for improving human factor skills in multidisciplinary teams. In situ simulation can increase accessibility and promote ward team learning, but its effectiveness with respect to frailty has not been explored. Method: A single-centre, multi-fidelity, inter-professional in situ frailty simulation programme was developed. One-hour sessions were delivered weekly using frailty-based clinical scenarios. Mixed-method evaluation was used, with data collected pre- and post-session for comparison. Results: In total, 86 multidisciplinary participants attended 19 sessions. There were significant improvements in self-efficacy rating across 10 of 12 human factor domains and in all frailty domains (*p* < 0.05). The common learning themes were situational awareness, communication and teamwork. Participants commented on the value of learning within ward teams and having the opportunity to debrief. Conclusion: In situ simulation can improve the self-efficacy of clinical and human factor skills related to frailty. The results are limited by the nature of self-reporting methods, and further studies assessing behavioural change and clinical outcomes are warranted.

## 1. Introduction

The prevalence of frailty among those aged over 50 in England was 8.1% in 2020, with a sharp increase in prevalence observed with advancing age [1]. An ageing population means that this prevalence of frailty is only likely to increase further [2]. Frailty is also prevalent globally. A study in 2020 across 62 countries worldwide reported a prevalence of 12% using physical frailty measures and 24% when using a deficit accumulation model [3].

Older adults living with frailty account for over 4000 hospital admissions daily in the UK [4], constituting 30% of patients in acute medical units [5] and up to 50% of inpatients [6]. Once in a hospital, they are more likely to experience adverse events, fall, develop delirium and suffer due to medication errors [7,8]. The presence of frailty increases the average length of stay by more than 5 days, results in a 32% increase in the likelihood of functional decline and is associated with an overall mortality risk that is 3.49 times greater than for patients without frailty [9].

The understanding of frailty and how it presents is variable among hospital staff [8]. For example, 72% of delirium is potentially undetected or misdiagnosed [10], meaning longer hospital stays [11], higher rates of institutionalisation and higher mortality [12]. This unwarranted variation in the quality of care delivered to patients living with frailty has been recognised at a national level [4,8]. On a global scale, the impact of frailty is only predicted to increase as populations age, making it an area in need of urgent attention [13]. One way to focus on this is to address the identified gap in clinical skills in order to support healthcare professionals caring for patients with frailty and to improve patient safety, experience and outcomes.

The gold standard management for patients with frailty is a comprehensive geriatric assessment [14]. A comprehensive geriatric assessment requires a multidisciplinary approach and has been shown to improve outcomes in patients living with frailty following admission to a hospital [15]. Human factor skills—how people interact and function with other people and the environment around them—are vital for the effective functioning of a multidisciplinary team (MDT) [16]. Educational initiatives ought to be developed with this in mind, promoting effective multidisciplinary collaboration and addressing the needs of the entire MDT.

Simulation is an established educational technique used to recreate ‘real life’ experiences in an immersive manner to facilitate learning and improve patient care [17]. It is underpinned by Kolb’s Experimental Learning Theory [18], a four-stage learning cycle whereby learners undergo a *concrete experience* via a simulated scenario, followed by *reflective observation* on the experience (such as during debriefing), leading to *abstract conceptualisation* where the relevance of and possible new approaches to the situation are considered, before these are implemented in future practice via *active experimentation* [19]. This not only offers the opportunity to develop new knowledge and understanding but also critically challenges, via reflection, pre-existing assumptions and beliefs, which can have a wider influence on the subsequent behaviours of learners [20].

When undertaken within an inter-professional group, this can allow for sharing across professional boundaries to provide a greater understanding of other professional roles and the development of the skills needed for effective inter-professional working, improving inter-professional collaboration and patient care [19,21,22]. Using simulation is also an advantageous way of delivering inter-professional education as it can be structured to be relevant for all learners, amalgamates knowledge within a team and offers the opportunity for debriefing, which may reveal existing social hierarchy, diversity and divisions [23]. This makes simulation a favourable option for addressing the identified educational gap for staff caring for patients with frailty whilst meeting the needs of the entire MDT.

In-centre simulation, occurring in specialist simulation centres away from the clinical environment, has been used to effectively deliver inter-professional education in geriatric medicine [24]. In situ simulation, occurring within the clinical environment, can provide increased accessibility to training in addition to promoting intact ward team learning and highlighting potential latent safety issues [25,26]. It has been used successfully in acute medicine and intensive care, and its incorporation is recommended by the Royal College of Physicians to facilitate learning in a busy work environment [27]. Its use within geriatric medicine has only occurred on a small scale [28], addressing areas such as delirium [29]. This paper reports the development and evaluation of a frailty-specific, multidisciplinary in situ simulation programme aimed at improving clinical and human factor skills for staff that care for patients with frailty.

## 2. Method

A multidisciplinary working group including geriatricians, simulation faculty and allied healthcare professionals developed the programme and learning outcomes. The learning outcomes of the programme were aligned with the British Geriatric Society’s “Frailty Hub” [30] and Health Education England’s “Frailty Framework of Core Capabilities” [31]. These are outlined below:Demonstrate the ability to recognise a frail patient, including common clinical presentations associated with frailty;Demonstrate the skills required to manage patients with frailty, including the management of frailty syndromes and acute deterioration in patients with frailty;Describe the possible adverse outcomes of frailty in an inpatient environment;To show an understanding of the limitations of treatment in frail patients and advanced care planning;To illustrate an understanding of human factor skills and their impact on clinical performance, particularly within the multidisciplinary team.

A bank of simulated scenarios was created with input from across the MDT, and each training session would focus on one of these scenarios. The scenarios involved a simulated patient living with frailty as defined as a state of vulnerability relative to a poor resolution of homoeostasis after a stressor event [32]. All simulated patients used within the scenarios had a Rockwood Clinical Frailty Scale (CFS) [33] of 5 or more (at least mildly frail). The CFS of the simulated patients differed between each scenario (ranging 5–8) to give participants experience of managing varying degrees of frailty. A list of the different simulated scenarios used is outlined below:Hypoactive delirium due to urinary retention and constipationFall with a fractured neck of femurFall with a head injury on anticoagulationFamily discussion following a fallSeizure due to missed anti-epileptic medication in hypoactive deliriumAspiration pneumonia in advanced frailty requiring treatment escalation decision making and discussion with familyAcute strokeIatrogenic fluid overload in advanced frailty

A multidisciplinary faculty ran the simulation and led the debrief. Sessions lasted for one hour comprising an introduction, simulated scenario and debrief. The programme was delivered on a weekly basis across the Older Persons’ Unit (OPU) with participants including nurses, doctors and allied healthcare professionals who were all informed in advance alongside liaison with the ward manager.

Sessions were open to all staff across the OPU (including students) and advertised by the leads within each profession. Professional groups who only attend the OPU through referral, rather than being attached to the unit, were not included. Staff wanting to attend were allocated sessions on a rotational basis to maximise accessibility to sessions across the ward teams. Attendance was not mandatory, and staff could attend multiple sessions. Non-clinical staff were excluded as scenarios were clinically focused.

A multi-fidelity approach was taken to scenarios with both high- and low-fidelity manikins, alongside simulated patient actors. Scenarios ran with 3–6 participants to ensure the numbers were sufficient for the sessions without overcrowding or impeding participants’ ability to become involved in the scenario. These took place in patient bays, patient side rooms or ward day rooms as relevant to the scenario and to accommodate bed availability.

Pre-session information was sent to participants via email, and participants were asked to complete pre- and post-session questionnaires (the original data is available in the Appendix A). These questionnaires were semi-structured with self-efficacy-rated questions for human factor skills using the Human Factors Skills for Healthcare Instrument (HuFSHI), a validated tool for the evaluation of the self-efficacy of human factor skills [34], and frailty clinical-based questions. Participants were asked to rate these from 1 (definitely cannot do) to 10 (definitely can do). Free-text responses to questions regarding the assessment of learning also formed part of the post-session questionnaire (Appendix B). Anonymous identifiers were used to pair pre- and post-session questionnaires. All participants consented to the use of their anonymised feedback data for the evaluation of the programme.

Statistical analyses were performed using Stata (StataCorp. 2021. Stata Statistical Software: Release 17. College Station, TX, USA: StataCorp LLC.). Paired scores from the questionnaires completed before and after training were compared using Wilcoxon signed-rank test for non-parametric data for testing the hypothesis of no difference between the groups.

Data from the free-text questions were thematically analysed by author L.D. via the process of iteratively reading, grouping and then re-grouping data into common themes. This was checked by authors G.W. and A.N.B., with any discrepancies discussed and re-grouped if needed. Single responses could be coded against multiple themes.

The development and evaluation of this project were registered with and adhered to the standards of the Clinical Governance Department at our institution (Service Evaluation Project Number 11484) and did not require submission to a research ethics committee.

## 3. Results

In total, 19 in situ simulation training sessions were delivered between October 2020 and September 2021, with a total of 86 participants. Participants spanned the MDT and included the following: doctors (31), nurses (28), nursing associates (6), nursing assistants (5), nursing students (2), physiotherapists (7), physiotherapy assistants (3), physiotherapy students (3) and occupational therapists (1).

Paired pre- and post-session self-efficacy scores were available for 63 participants. Quantitative data show significant improvements in self-efficacy ratings in 10 out of 12 HuFSHI questions and in all frailty domains (Table 1).

Course participants’ free-text comments relative to post-session questions yielded 308 individual responses (204 about what was learnt and 108 about what they felt was good about in situ training), which were coded into 8 key themes (Table 2).

All participants reported that the sessions were useful but areas for improvement included the potential for longer training sessions and a better training environment. Notably, some latent threats were identified during sessions, including the lack of access to key emergency equipment for some staff and shared access to some important equipment across wards. These identified latent threats were then fed back to the department governance lead and used to inform ongoing quality improvement work.

## 4. Discussion

Simulation is recognised as being an effective teaching modality for developing a variety of clinical and human factor skills for healthcare professionals managing a complex and challenging older population [24]. The COVID-19 pandemic has led to disruption in education and training [35,36,37], with high levels of staff sickness and isolation [38], making staff release for training a challenge in addition to reducing in-centre capacities due to social distancing. Implementing this in situ frailty simulation project has allowed valuable staff training to continue despite these challenges, with evaluation showing that it improved clinical and human factor skills across the MDT. Whilst created in response to the pandemic, this service evaluation illuminates the likely efficacy of this educational project outside of this context and its potential beneficial role in increasing access to training.

The strengths of this programme include the number of participants from a range of professional healthcare backgrounds and the mixed method evaluation of participant involvement. The use of a validated tool (HuFSHI) to evaluate self-efficacy provided an evidence-based and quantitative way of measuring participants’ learning. The use of a bank of simulation scenarios also increased the variety of learning and allowed flexibility when bed spaces were not available. In situ simulation removes the reliance on simulation centre availability, meaning that sessions can be delivered in departments with limited access to a simulation centre. The variety of simulation modalities used (high- and low-fidelity simulated patient manikins, alongside simulated patient actors) suggests that departments would not need extensive access to equipment, potentially lowering costs and demand on resources [26]. Sessions had the benefit of being ward-team-based, allowing intact ward teams to work and learn together in their normal clinical environment [26,39]—this was echoed in feedback from participants and is in keeping with previous studies [19,22]. There are challenges when using in situ simulation, most noticeably the lack of bed spaces in which to run scenarios. Having scenarios that can be run in other locations (such as the day room) meant that sessions were never cancelled due to a lack of bed space.

The limitations of the project include the nature of self-reporting by participants, which raises questions as to whether learning from sessions will be transferred into practice. Self-efficacy, however, has been positively and strongly linked to work-related performance [40], suggesting that improvements may well translate into practice. Another limitation is the fact that the scenarios used were not standardised for each session. We were limited in the number of participants attending sessions to ensure involvement in the scenario, although it would be possible to have observers of the scenario as learning still occurs in the observer role during simulation [41]. The effectiveness of the different simulation modalities was not measured against each other and warrants further exploration.

Future studies on the role of simulation in frailty-based education would be beneficial—particularly looking at the implementation of in situ simulation in different centres with different resources, levels of fidelity and scenarios. In addition to using a control group that did not attend the simulation and follow-up questionnaires to assess the retention of knowledge and skills. It would also be of value to look at the impact of sessions on staff behaviour and patient outcomes.

This project has demonstrated that in situ simulation can improve the self-efficacy of clinical and human factor skills related to frailty. Challenges for delivery include time pressures and bed space, but having different scenario options and faculty adaptability can overcome this even within the constraints of a busy ward environment. In situ simulation is a feasible and effective way of delivering multidisciplinary education with respect to complex geriatric topics and can improve the human factor skills crucial for frailty management.

## 5. Conclusions

There is a recognised gap in knowledge and clinical skills that exists for staff caring for patients with frailty. In situ frailty-based simulation presents a solution for clinical teams in addressing this gap. Simulation provides a way of bringing the MDT into the educational experience of staff, allowing intact teams the opportunity to learn and improve together. The mode of delivery also abates the increasing challenge of clinical demands and staffing pressure. Although there is a positive effect on the self-efficacy of participants, further studies are necessary in order to confirm that this translates into staff knowledge and behaviour and, ultimately, patient outcomes. Additional work examining the impact of different simulated scenarios and varying levels of fidelity on learning would also be of value.

## Figures and Tables

**Table 1 geriatrics-08-00026-t001:** Pre- and post-session HuFSHI and frailty domain self-efficacy ratings.

	Median Score (IQR)		*p* Value
	Pre-Session	Post-Session	
**HuFSHI Domain**			
Constructively managing others’ negative emotions at work	7 (5–8)	8 (6–8.5)	<0.001
Requesting help from colleagues in other professions	8 (7–10)	8 (8–9)	0.266
Communicating effectively with a colleague with whom you disagree	7 (5–8)	8 (7–8.5)	<0.001
Prioritising when many things are happening at once	7 (6–8)	8 (7–9)	<0.001
Speaking up as part of a team to convey what you think is going on	7 (6–8.5)	8 (7–9)	<0.001
Involving colleagues in your decision making process	8 (7–9)	8 (7.5–9)	0.871
Dealing with uncertainty in your decision making process	7 (5.5–8)	8 (7–9)	<0.001
Asking other team members for the information I need during busy ward environment	8 (7–9)	8 (7.5–9)	0.001
Recognising when you should take on a leadership role	7 (6–8)	8 (7–9)	<0.001
Monitoring the ‘big picture’ during a complex clinical situation	7 (5–8)	8 (7–9)	<0.001
Anticipating what will happen next in clinical situations	6 (5–8)	8 (7–8)	<0.001
Working effectively with a new team in clinical situations	7 (6–9)	8 (7–9)	<0.001
**Frailty-based domains:**			
Recognising a frail patient	8 (7–8.5)	8 (7–9)	0.007
Assessing a frail patient	7 (6–8)	8 (7–8.5)	<0.001
Recognising deterioration in a frail patient	7 (6–8)	8 (7–9)	<0.001
Anticipating potential adverse outcomes of a frail patient in hospital	7 (6–8)	8 (7–9)	<0.001
Managing a patient who has had a fall	7 (6–9)	8 (7–9)	<0.001
Managing a patient with immobility	8 (6.5–9)	8 (7–9)	0.002
Managing a patient with delirium	7 (6–8)	8 (7–9)	<0.001
Managing a patient with incontinence	8 (5–9)	8 (6.5–9)	0.001
Recognising a patient’s susceptibility to side effects of medication	7 (5–8)	8 (6–9)	<0.001

**Table 2 geriatrics-08-00026-t002:** Thematic analysis of free-text responses to questions about the assessment of learning from in situ simulation training sessions with representative comments from participants.

What Have You Learn from Today’s Session?	Comments
Theme 1.1 Teamwork	“To use the competencies in the team to their best potential” (ST) “Better understanding of MDT roles” (Student PT) “Using the mental model out loud as this helps everyone” (Band 5 SN)
Theme 1.2 Communication	“Closed loop feedback” (Clinical fellow) “CUS [Concerned, Uncomfortable, Safety] communication tool” (OT) “To ask for clarity if I’m not sure how to help” (Band 6 PT)
Theme 1.3 Situational and spatial awareness	“Awareness of surroundings” (Assistant Practitioner) “Understanding the bigger picture” (FY1) “Understand the complexity in each situation” (Band 6 SN)
Theme 1.4 Clinical knowledge	“Identifying frail patients” (Band 5 SN) “Scoop Team, don’t hoist [for fall with possible neck of femur fracture]” (NA) “Thinking about ceilings of care in a deteriorating patients” (FY1)
**What was good about today’s session?**	**Comments**
Theme 2.1 Session were team based	“Helped remind me that other team members may not feel competent in certain scenarios and it helps to just be sensitive of this and work as a team” (ST) “How everyone worked together” (Student nurse) “Get to know members of the ward and MDT as a new member of staff” (Band 6 PT)
Theme 2.2 Use of discussion and debrief	“Helpful to debrief afterwards” (FY1) “Open environment feeling like I could speak up” (Assistant PT) “Everything explained well in debrief” (Student nurse)
Theme 2.3 Learner improvement in knowledge or skills	“Lots of learning points to help with similar scenarios moving forwards” (Senior House Officer) “Making staff to be on point with care delivery” (Band 5 SN) “Critical thinking, insight into managing a trauma patient” (Senior NA)
Theme 2.4 Using in-situ simulation	“In situ was actually on a ward environment which adds to realism” (Unknown) “Simulating events that often happen on the ward was helpful” (PT) “Good to practice caring for a deteriorating patient in a safe environment” (Band 6 SN)

FY1—foundation year 1 doctor; ST—specialist trainee doctor; SN—staff nurse; OT—occupational therapist; PT—physiotherapist; NA—nursing associate.

## Data Availability

The data presented in this study are available in the Appendix A.

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
