# Peer review of "In Situ Simulation Training for Frailty"

_geriatrics, 2023, doi:10.3390/geriatrics8010026_

Round 1

Reviewer 1 Report

Frailty is a syndrome that requires specific screening or diagnostic scales to assess it.

Dent E, Kowal P, Hoogendijk EO (2016). Frailty measurement in research and clinical practice: A review. European Journal of Internal Medicine, 1:3-10. doi:10.1016/j.ejim.2016.03.007

The authors do not include any validated scale in their methods, and it seems that they use the term frailty as something subjective that the doctor evaluates. Therefore, the study from the methodological point of view is deficient and does not show that it is based on frailty.

On the other hand, the authors refer in the same introduction that it has already been shown that simulation training is effective for geriatric assessment in hospitalized patients, therefore, by not demonstrating that they measure frailty, it seems to me that the article is not very relevant as it is currently proposed.

Author Response

We thank you for your comments and have taken them into consideration in the revised manuscript. Please see the attached cover letter to the academic editor for more detail.

Reviewer 2 Report

Dear Editor,

The Manuscript entitled "In-Situ Simulation Training For Frailtys" by Liam Dunnell in on the a single centre, multi-fidelity, inter-professional in-situ frailty simulation program on 86 participants spanning the multidisciplinary team attended 19 sessions. There were significant improvements in self-efficacy rating across 10 of 12 human factors domains and all frailty domains. The Authors demonstrated that in-situ simulation is an effective way of delivering training to staff caring for frail patient.

Showing their data with the evidence of the literature r strengthens their explanation and must contribute to our knowledge.

However, in this article, there are several misuses of technical terms/misinformation.

The title reflects the main subject of the study.
The abstract summarizes and reflect the work described.

Minor comments:

Extend introduction section

Add conclusion section  to describe 'other data' required and provide further details on recommendations for physicians based on the examined studies.

Table 1: List of simulation scenarios. better modulate graphically

Table 2. Pre- and post-session HuFSHI and frailty domain self-efficacy ratings. better modulate graphically

Table 3. better modulate graphically

Include details on how patients should be monitored for symptoms.

Kind regards

Author Response

We thank you for your comments and have taken them into consideration in the revised manuscript. In particular, we have extended the introduction section and added a conclusion. Please see the attached cover letter for details of all the changes made in the revised manuscript.

Reviewer 3 Report

I have reviewed the manuscript "In-Situ Simulation Training For Frailty", which regards a relevant topic for the journal and the geriatric field. It provides useful data, it is well-written, easy to follow and the results are clearly presented. My only concern is that the design of the study deals with an important limitation due to the self-reporting by participants. The authors acknowledge this in the limitations paragraph, yet because of the relevance of this issue, I would advice to add mentioning of this in the abstract. Thus, it will be easier for the reader to identify this limitation from the begining, which improves interpretation of the results and potential application for clinical purposes. 

Author Response

We thank you for your comments and have taken them into consideration in the revised manuscript. In particular, we have included the important limitation of self-reporting by participants in the abstract. Please see the attached cover letter to the academic editor for details of all of the changes made in the revised version of our manuscript.

Round 2

Reviewer 1 Report

The authors do not respond to or correct my main comment.

"Frailty is a syndrome that requires specific screening or diagnostic scales for its evaluation.

Dent E, Kowal P, Hoogendijk EO (2016). Measurement of frailty in research and clinical practice: a review. European Journal of Internal Medicine, 1:3-10. doi:10.1016/j.ejim.2016.03.007

The authors do not include any validated scale in their methods, and it seems that they use the term frailty as something subjective that is assessed by the clinician.

The authors do not include any validated scale in their methods, and it seems that they use the term frailty as something subjective that is assessed by the clinician.

 Therefore, the study from a methodological point of view is deficient and does not demonstrate that it is based on frailty."

I suggest to the authors that they could change the focus of the article or that they could show that it is based on frailty by specifying how they measure it. At this time, what they really seem to be evaluating is a comprehensive geriatric assessment, not frailty.

Author Response

Thank you for your feedback.  Please see our revised manuscript. The methods section has been adjusted and changes highlighted . The scenarios were all developed around patients living with frailty. The frailty scale we use in to develop these scenarios is the Rockwood Clinical Frailty Scale. All the scenarios were designed around patients with a CFS of 5 or more (at least mildly frail). The frailty based competencies we measure are  aligned with the British Geriatric Societys Frailty Hub” and Health Education England’s Frailty Framework of Core Capabilities” and are nationally recognised required competencies for professionals caring for patients living with frailty.